# Microbiological Characterization of *Cutibacterium acnes* Strains Isolated from Prosthetic Joint Infections

**DOI:** 10.3390/antibiotics11091260

**Published:** 2022-09-16

**Authors:** Llanos Salar-Vidal, John Jairo Aguilera-Correa, Holger Brüggemann, Yvonne Achermann, Jaime Esteban

**Affiliations:** 1Clinical Microbiology Department, IIS-Fundación Jiménez Díaz, 28020 Madrid, Spain; 2CIBER de EnfermedadesInfecciosas (CIBERINFEC), 28020 Madrid, Spain; 3Department of Biomedicine, Aarhus University, 8000 Aarhus, Denmark; 4Division of Infectious Diseases and Hospital Epidemiology, University Hospital of Zurich (USZ), 8091 Zurich, Switzerland; 5Faculty of Medicine, University of Zurich (UZH), 8091 Zurich, Switzerland; 6Internal Medicine, Hospital Zollikerberg, 8091 Zurich, Switzerland

**Keywords:** *Cutibacteriumacnes*, prosthetic joint infection, biofilm, antimicrobial susceptibility, phylotyping

## Abstract

Aims: This study aimed to characterize 79 *Cutibacterium acnes* strains isolated from prosthetic joint infections (PJIs) originated from eight European hospitals. Methods: Isolates were phylotyped according to the single-locus sequence typing (SLST) scheme. We evaluated the ability of the biofilm formation of *C. acnes* strains isolated from PJIs and 84 isolates recovered from healthy skin. Antibiotic susceptibility testing of planktonic and biofilm cells of PJI isolates and skin isolates was performed. Results: Most of the isolates from PJIs belonged to the SLST class H/phylotype IB (34.2%), followed by class D/phylotype IA_1_ (21.5%), class A/phylotype IA_1_ (18.9%), and class K/phylotype II (13.9%). All tested isolates were biofilm producers; no difference in biofilm formation was observed between the healthy skin group and the PJI group of strains. Planktonic and sessile cells of *C. acnes* remained highly susceptible to a broad spectrum of antibiotics, including beta-lactams, clindamycin, fluoroquinolones, linezolid, rifampin, and vancomycin. The minimal inhibitory concentrations (MICs) for planktonic and biofilm states coincided in most cases. However, the minimal biofilm eradication concentration (MBEC) was high for all antimicrobial drugs tested (>32 mg/L), except for rifampin (2 mg/L). Conclusions: *C. acnes* strains isolated from healthy skin were able to produce biofilm to the same extent as isolates recovered from PJIs. All *C. acnes* strains in planktonic and sessile states were susceptible to most antibiotics commonly used for PJI treatment, although rifampin was the only antimicrobial agent able to eradicate *C. acnes* embedded in biofilm.

## 1. Introduction 

Prosthetic joint infections (PJIs) are one of the most-feared complications after joint replacement surgery, as they increase the morbidity, mortality, and costs of the health care system [1]. PJIs are difficult to treat and frequently require surgical intervention and prolonged antibiotic therapy [2].

*Cutibacterium acnes* is a common anaerobic bacterium isolated from PJIs [3], especially associated with shoulder infections [4]. Albeit having an overall low virulence, the pathogenicity of this microorganism involves a few virulence traits, including biofilm formation, a trait that is highly relevant in implant infections [5]. There is growing evidence that in vivo biofilm formation might play a key role in antibiotic treatment failure [6].

Based on molecular techniques, the *C. acnes* population has been divided into six main phylotypes (IA1, IA2, IB, IC, II, and III) [7,8,9]. Each phylotype has been associated with different types of infections and healthy skin, respectively [9,10,11,12,13]. In the skin, the predominant phylotype is IA_1_; this type, together with IA_2_, is also the most frequent type associated with moderate to severe acne. Phylotypes IB and II are more frequently involved in bloodstream-, soft tissue-, and implant-associated infections, whereas type III, frequently isolated from the skin of the lower back has been associated with spinal disc infections and progressive macular hypomelanosis [9,10,11,12,13,14]. Recently, a single-locus sequencing typing (SLST) scheme with high resolution has been developed for the typing of *C. acnes* strains beyond the phylotype level [11]. Ten main SLST classes are distinguished: A, B, C, D, E, F, G, H, K, and L. Phylotype IA_1_ comprises SLST classes A-E, whereas phylotypes IA_2_, IC, IB, II, and III correspond to SLST classes F, G, H, K, and L, respectively [11].

This study aimed to characterize 79 *C. acnes* strains from PJIs; the strains originated from eight European hospitals. We evaluated the biofilm formation of *C. acnes* isolates obtained from PJIs and compared it with isolates recovered from healthy skin. In addition, all PJI *C. acnes* strains were typed according to the SLST scheme and susceptibility testing of planktonic and biofilm-embedded strains for different antimicrobials was performed. 

## 2. Results

The clinical data of *C. acnes* strains isolated from PJI are shown in Appendix A. The results in relation to biofilm formation are shown in Table 1. In our cohort, most of the isolates were recovered from hip PJIs (50.6%), followed by shoulder PJIs (34.2%) and knee PJIs (12.6%). Regarding the assignment to SLST classes, the most predominant one was class H/phylotype IB (34.2%), followed by class D/phylotype IA1 (21.5%), class A/phylotype IA1 (19%), and class K/phylotype II (13.9%). Regarding the joint site of infection, class H/phylotype IB was the most common SLST class in hip (15/40) and shoulder (9/27) PJIs, whereas for knee PJIs, class D/phylotype IA1was prevalent (4/10).

The biofilm formation of 79 isolates recovered from PJIs was compared with 84 *C. acnes* isolates from healthy skin (HS). According to the methodology used, all the isolates were biofilm producers. In the PJI group, more than half of the isolates were moderate biofilm producers (50.6%), followed by weak biofilm producers (32.9%) and strong biofilm producers (16.5%). In the HS group, the majority of the isolates were moderate biofilm producers (57.1%), followed by weak biofilm producers (35.8%) and strong biofilm producers (7.1%). Interestingly, the proportion of strong biofilm producers was significantly higher in the PJI group than in the HS group (*p*-value = 0.035).

Neither biofilm formation according to the origin of the sample, nor gender, nor joint site of infection, nor immunosuppression of the patient showed statistically significant differences (*p*-value > 0.05). Only a significant difference was detected regarding the SLST class (*p*-value < 0.05). SLST class F/phylotype IA_2_ was a significantly stronger biofilm producer compared with the other SLST classes (*p*-value < 0.05); however, only three strains of SLST class F were examined here. There was no correlation between the ability to form biofilm and the patients’ age (Spearman’s ρ = 0.974; *p* value = 0.3933).

The antibiotic susceptibility results are shown in Table 2. Planktonic cells of *C. acnes* were highly susceptible to penicillin, amoxicillin-clavulanic acid, clindamycin (only three resistant isolates), levofloxacin (only two resistant isolates), linezolid, rifampin, and vancomycin. In most of the cases, MBC was slightly superior to MIC. The MBC for clindamycin and rifampin was equal to MIC. Except for levofloxacin, all antimicrobial drugs tested showed notable bactericidal capacity against the planktonic form of *C. acnes*.

MBIC values were in accordance with MIC values, showing that the susceptibility of *C. acnes* embedded in biofilm is comparable to its planktonic form. MBEC values were above 32 mg/L, except for rifampin (0.5 mg/L), showing that rifampin was the only one with the potential to eradicatebiofilm-embedded *C. acnes*.

## 3. Discussion

In this study, we determined the phylogenetic types, biofilm formation capacities, and susceptibility profiles of *C. acnes* strains isolated from PJIs in eight European hospitals. 

The molecular typing of *C. acnes* isolates from implant-associated infections was performed to establish an association between certain phylotypes and infection. In our cohort, the SLST classH/phylotype IB was most frequently detected, which is in accordance with previous studies [15]. The other two prevalent SLST classes, A and D, both belong to the phylotype IA_1_. Some studies have reported phylotype IA_1_, in particular SLST class A, as predominant in their cohort studies [16,17,18]. Most skin microbiome studies have shown that phylotype IA_1_ strains are most dominant on healthy human skin; again, SLST classA dominates [11,13]. Thus, it is currently not clear if a certain SLST classor phylotype has a higher PJI-causing potential; discrepancies between different studies might be due to the types of joints involved, types of infection (monomicrobial or polymicrobial), or geographical origin.

Regarding the joint site of infection, no association with specific phylotypes or SLST classes was found, in contrast to a previous study [18] that reported a higher rate of SLST classD in shoulder and knee PJIs compared to hip PJIs. 

One of the main virulence properties associated with PJIs is biofilm formation [1]. Several studies have proven the ability of *C. acnes* to develop a biofilm both in vitro and in vivo [6,19]. In our study, all isolates tested (healthy skin and PJI isolates) were biofilm producers, most of them being moderate producers (88/163); our results detected a slightly higher rate of strong biofilm producers in PJI isolates. This is largely in agreement with the study of Holmberg et al. [20] that did not show significant differences ofthe biofilm formation capacity of *C. acnes* isolates from patients with PJIs and healthy skin. These resultsindicate that *C. acnes* strains isolated from healthy skin have the potential to cause an invasive infection associated with biofilm formation. 

No differences between biofilm formation and gender, type of implant, immunosuppression of the patient, or phylotype were observed. In contrast, Kuehnast et al. [21] reported that biofilm formation differed between *C. acnes* phylotypes, rather than the anatomical site of isolation. Other studies showed that SLST class A had the strongestability toproduce biofilm in the microtiter plate assay [22]. These discrepancies might be related to the methodology used, number of isolates, geographical origin, and cohort composition, since not all SLST classes are equally represented in thepublished studies. 

An optimal treatment for PJIs caused by *C. acnes* has not yet been established. This bacterium is usually susceptible to a wide range of antibiotics, such as beta-lactams, fluoroquinolones, rifampin, and vancomycin [23]. The in vitro susceptibility results of planktonic cells are congruent with previous studies and also with studies that evaluated *C. acnes* strains isolated from PJIs [22,23]. In our cohort, we found that almost all the isolates were susceptible to all antimicrobial agents tested. Interestingly, according to MIC and MBIC, biofilm cells were not more resistant than planktonic cells, as has been reported for *Staphylococcus epidermidis* isolates [24]. Nevertheless, significant differences were found in terms of bactericidal concentration and biofilm eradication concentration. Rifampin, beta-lactams, clindamycin, and vancomycin showed a high bactericidal effect. However, all antibiotics failed to eradicate *C. acnes* biofilm within the recommended dosing ranges, as previously described [25], with the exception of rifampin.

Antibiotic regimensto treat PJIs caused by *C. ac**nes* often include rifampin, because it has been demonstrated in PJIs that this antimicrobial drug has the potential to penetrate the biofilm and is effective against the stationary bacteria of the biofilm [26]. In our study, rifampin was the antibiotic that showed the highest activity against both planktonic and biofilm states of *C. acnes*, supporting the use of rifampin in the treatment of *C. acnes* PJIs. This is in line with a multicenter European study that included 187 patients with *Cutibacterium* spp. PJI [27]. The study showed tentative evidence for a beneficial effect of adding rifampin to the antibiotic treatment with decreased hazards for developing treatment failure, though not statistically significant.

Our study has a few limitations. A limited number of isolates were included, due to the fact that *C. acnes* is an uncommon pathogen in PJIs, and thus not all SLST classes/phylotypes are equally represented. In addition, the biofilm assays were performed using plastic surface-based procedures in rich culture media. Such conditions are not comparable with in vivo scenarios with implant surfaces and the scarce availability of nutrients for bacterial proliferation that are derived from the human host. Biofilm assays that better mimic the in vivo scenario are needed.

## 4. Methods

### 4.1. Strain Isolation and Identification

Seventy-nine *C. acnes* strains recovered from PJIs from eight European hospitals as part of a multicenter study (Kusejko et al. [27]) were used. Inclusion and exclusion criteria used in this study were based on the study of Kusejko et al. [15]. Unfortunately, because this is a retrospective study, not all the strains were available. In addition, eighty-four isolates from healthy skin of 84 volunteers unrelated to the health care system were collectedusing cottonswabs from alar and retroauricular creases, due to *C. acnes* high relative abundance in those areas [28]. Isolates from PJIs and healthy skin were recultured onto Schaedler-5% sheep blood agar plates (bioMérieux, Marcy l’Étoile, France) for 48 h at 37 °C under anaerobic conditions and were identified by MALDI-TOF MS (Vitek MS, bioMérieux, Marcy l’Étoile, France). 

### 4.2. Typing of PJI C. acnes Strains

SLST typing was performed on PJI isolates. After DNA extraction, the SLST fragment was amplified by PCR using the primers 5′-CGCCATCAAGGCACCAACAA-3′ and 5′-ATATCGGCCCGTATTTGGGC-3′. Amplification was achieved starting with a cycle of 40 s of denaturation at 96 °C, followed by 35 cycles of 35 s at 94 °C, 55 °C for 40 s, 72 °C for 40 s, and a final step for extension at 72 °C for 7 min. Electrophoresis in agarose gel was performed to verify the amplification. Purified products were Sanger-sequenced from both directions (Eurofins Genomics, Galten, Denmark). SLST classes were assigned using the SLST allele database (http://medbac.dk/slst/pacnes, accessed on 17 August 2022).

### 4.3. Biofilm Formation

Biofilm formation was evaluated using a modified method of Stepanovic et al. [29]. Eight wells were filled with 200 µL of brain heart infusion (BHI) broth supplemented with 2% glucose. A bacterial inoculum of 10^6^colony-forming units (CFUs) per mL per well were statically incubated at 37 °C under anaerobic conditions in treated flat-bottomed 96-well plates (Falcon^®^, Thermo Fisher Scientific, Boston, MA, USA). After 72 h incubation, the broth was removed, and wells were rinsed two times with methanol (PanReac AppliChem, Chicago, IL, USA), and 150 µL of 1% crystal violet was used for staining. The optical density (OD) of each well was measured at 570 nm using a TECAN Infinite 200 microtiter platereader (Tecan Group Ltd., Männedorf, Switzerland). Each isolate was tested in triplicate (*n* = 24 per strain). Strains were divided into different categories based on the OD values, and the following cut-off values (optical density control (ODc)) were established: no biofilmproducer, OD ≤ ODc; weak biofilmproducer, ODc < OD ≤ 2×ODc; moderate biofilm-producer, 2×ODc < OD ≤ 4×ODc; strong biofilmproducer, 4×ODc < OD.

### 4.4. Susceptibility Testing

All PJI strains were tested for antibiotic susceptibility to amoxicillin-clavulanate, clindamycin, levofloxacin, linezolid, penicillin, rifampin, and vancomycin.

The minimal inhibitory concentration (MIC) and the minimal bactericidal concentration (MBC) were evaluated according to the broth microdilution method described by the European Committee on Antimicrobial Susceptibility Testing (EUCAST) [30]. Colonies of the isolates were suspended in Mueller–Hinton cation-adjusted broth. Sterile round-bottomed 96-well plates were inoculated with 100 µL of Mueller–Hinton cation-adjusted broth containing the antimicrobial agent plus 100 µL of the bacterial suspension for obtaining a final inoculum of 10^4^ CFU per well and incubated under anaerobic conditions. After 48 h of incubation, the MIC was determined at the first antibiotic concentration where there was no turbidity. Subsequently, 50 µL of each well was transferred into a new plate containing 150 µL of Mueller–Hinton cation-adjusted broth by using a modified flash microbicide method [31]. Plates were incubated for 2 days, and the MBC was determined at the first antibiotic concentration where there was no bacterial growth.

The minimal biofilm inhibitory concentration (MBIC) and the minimal biofilm eradication concentration (MBEC) were assessed following the protocol described by Coenye et al. [32]. Briefly, colonies of the isolates were transferred into sterile phosphate-buffered saline (PBS) (Sigma-Aldrich, Burlington, MO, USA), and the supernatant was adjusted to a turbidity of 0.5 ± 0.02 McFarland. Sterile polystyrene-treated flat-bottomed 96-well plates were inoculated with 200 µL of the solution and incubated at 37 °C for 4 h in anaerobiosis jars (Oxoid Ltd., Thermos Fisher Scientific, Boston, MA, USA) for favoring the bacterial adhesion to the well bottom. After 4 h, the supernatant was removed, and each well was washed with 200 µL of sterile PBS. Thereafter, 200 µL of clostridial nutrient medium (CNM) (Sigma-Aldrich, Burlington, MO, USA) was added to each well. After 24 h of incubation under anaerobic conditions, the supernatant was again removed, and the plates were washed with 200 µL of sterile PBS. One hundred µLof CNM plus 100 µL of CNM containing the corresponding antibiotic concentration was added, and plates were further incubated at 37 °C under anaerobic conditions for 48 h. After incubation, the MBIC was determined at the first antibiotic concentration where there was no turbidity. For determining MBEC, each well was washed with 200 µL of sterile PBS, filled with 200 µL of fresh CNM, and vigorously scrapped with sterile pipette tips and homogenized. The MBEC was determined after 48 h of incubation at the first antibiotic concentration where there was no bacterial growth.

The EUCASTresistance breakpointswere used to interpret antimicrobial susceptibility results [18]. Breakpoints for levofloxacin, linezolid, and rifampin breakpoints have not been determined regarding anaerobic Gram-positive bacteria; therefore, *Staphylococcus aureus* breakpoints were used.

### 4.5. Statistical Analysis

We performed statistical analysis by using Stata Statistical Software, Release 11 (StataCorp2009). The quantitative variables are summarized as median and interquartile range values. A Wilcoxon test for comparing quantitative variables was used; if the Wilcoxon test found statistically significant differences among more than two groups, a post hoc Dunn’s pairwise comparison with a Benjamini–Hochberg procedure was performed. The possible relationship between age and the biofilm formation capacity of *C. acnes* isolates was evaluated by means of Spearman’s rank correlation coefficient. We considered a level of statistical significance of *p*-value < 0.05 in all tests.

## 5. Conclusions

Our study provides additional insight into *C. acnes* PJIs, regarding typing, biofilm formation, and antibiotic susceptibility. This is the first multicenter, multinational study that evaluates a high number of *C. acnes* strains involved in PJIs using phenotypical and genotypical approaches. It also compares these strains with strains from healthy skin in terms of biofilm formation. It is important for surgical prevention strategies to consider that *C. acnes* strains isolated from normal skin also have the same ability to produce biofilm as strains recovered from PJIs, because the characteristics from the PJI isolates and healthy skin isolates were similar, and no specific properties (phenotypical or genotypical) were found that could be used for the determination of the pathogenicity of the strain. So, clinical significance of the isolates must be evaluated according to conventional methods previously described. *C. acnes* isolates were found to be susceptible to commonly used antimicrobials, although only rifampin was able to eradicate the biofilmin vitro. However, this in vitro result must be considered only as a possibility for the treatment of the patients because many other factors can influence the outcome of patients. A prospective randomized control trial is needed to investigate the role of rifampin to cure biofilm-associated *C. acnes* PJIs.

## Figures and Tables

**Table 1 antibiotics-11-01260-t001:** Biofilm Formation according to Origin of the Sample, Gender, Joint Site of Infection, Immunosuppression of the Patient, Strain Phylotype, and SLST Class.

	N	Biofilm Formation (n-Fold ODc) (IQR)	*p*-Value
Origin of the sample			0.5953
HS	84	2.47 (1.71–3.32)	
PJI	79	2.5 (1.78–3.44)	
Gender			0.2913
Male	60	2.54 (1.95–3.76)	
Female	19	2.50 (1.67–3,29)	
Joint site			0.4352
Elbow	1	2.98	
Hip	40	2.40 (1.74–3.52)	
Knee	10	2.90 (1.93–3.13)	
Shoulder	27	2.90 (1.83–3.77)	
Wrist	1	1.15	
Immunosuppression			0.7193
Yes	11	2.94 (2.27–3.29)	
No	64	2.53 (1.80–3.64)	
Phylotype			0.6427
IA	41	2.5 (1.77–3.40)	
IB	27	2.72 (1.78–3.53)	
II	11	2.49 (2.27–3.75)	
SLST			0.0449
A	15	2.38 (1.75–3.29)	
C	3	2.28 (1.41–2.50)	
D	17	2.97 (2.07–3.40)	
E	3	1.67 (1.30–2.18)	
F	3	4.96 (4.28–6.41)	
H	27	2.72 (1.78–3.53)	
K	11	2.49 (2.27–3.75)	

ODc: optical density control; IQR: interquartile range. Immunosuppression: see Appendix A.

**Table 2 antibiotics-11-01260-t002:** MIC_50_, MIC_90_, MBC_50_, MBC_90_, MBIC_50_, MBIC_90_, MBEC_50_, and MBEC_90_ for each Antibiotic of *C. acnes* Strains in mg/L.

Antibiotic	MIC_50_ (mg/L)	MIC_90_ (mg/L)	MBC_50_ (mg/L)	MBC_90_ (mg/L)	MBIC_50_ (mg/L)	MBIC_90_ (mg/L)	MBEC_50_ (mg/L)	MBEC_90_ (mg/L)
Amoxicillin-clavulanic acid	0.25	0.50	0.5	1	0.25	0.25	64	>256
Clindamycin	0.25	0.50	0.25	1	0.25	0.50	64	>256
Levofloxacin	0.50	1	1	2	0.50	1	>32	>32
Linezolid	0.50	1	1	2	0.50	0.50	>256	>256
Penicillin	0.0625	0.125	0.25	0.5	0.0625	0.125	32	> 32
Rifampin	0.03125	0.03125	0.03125	0.03125	0.03125	0.03125	0.50	2
Vancomycin	0.50	0.50	1	1	0.50	0.50	>256	>256

## Data Availability

Not applicable.

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
