# Peer review of "Microbiological Characterization of Cutibacterium acnes Strains Isolated from Prosthetic Joint Infections"

_antibiotics, 2022, doi:10.3390/antibiotics11091260_

Round 1
Reviewer 1 Report
Intresting paper for this special issue
please specify how the strains were selected from the work in CiD : (187 patients to 79 strains in the present work)
is the method used to evaluate biofilm formation able to detect strains that are not biofilm producers ? please specify it in the manuscript and/or provide a reference
In the method section line 213 specify that ODc is optical density control because it do not appera in the manuscript (just in the table)
Author Response
Response to Reviewer 1
- Please specify how the strains were selected from the work in CiD : (187 patients to 79 strains in the present work)
Unfortunately, It was impossible to collect all the strains that caused a PJI by C. acnes included in the previous study of Kusejko et al.
- Is the method used to evaluate biofilm formation able to detect strains that are not biofilm producers? please specify it in the manuscript and/or provide a reference
As it is indicated in the Methods section, not biofilm producers are detected when optical density was lower than the optical density control. The methodology described by Stepanovik et al. is a well-known technique used for the evaluation of biofilm production by different species of bacteria. We have referred it at the beginning of methods section.
- In the method section line 213 specify that ODc is optical density control because it do not appera in the manuscript (just in the table)
It has been changed in the manuscript.
Reviewer 2 Report
I commend the authors for their research entitled Microbiological characterization of Cutibacterium acnes strains isolated from prosthetic joint infections. The topic is interesting and the study is multicentric - an added value. My proposals are as follows.
Methods: This section is usually described after Introduction. Besides, please state clearly how the Cutibacterium acnes strains were obtained (e.g. using sonication?). Were Cutibacterium acnes strains the only bacteria isolated from samples? From which countries in Europe the strains were from?
Conclusion: This is the weakest part of the manuscript. What is making your study unique? How should – based on your findings - practicing orthopaedic surgeons change their algorithm in treating patients with PJI contaminated with Cutibacterium acnes strains? Please elaborate.
Author Response
Response to Reviewer 2
- Methods: This section is usually described after Introduction. Besides, please state clearly how the Cutibacterium acnes strains were obtained (e.g. using sonication?). Were Cutibacterium acnes strains the only bacteria isolated from samples? From which countries in Europe the strains were from?
Methods section appears at the end because we have used the template provided by the Journal.
Because of the retrospective nature of the study it is difficult to know the actual source of the used strains. However, the methodology used during the period of the study was described in the reference number 15, which is the clinical part of the multicenter project, and in all centers it was methodolody recommended in International guidelines.
Geographical origin of each of the strains used in the study is indicated in the Table S1 of the supplementary material.
- Conclusion: This is the weakest part of the manuscript. What is making your study unique? How should – based on your findings - practicing orthopaedic surgeons change their algorithm in treating patients with PJI contaminated with Cutibacterium acnes strains? Please elaborate.
We have modified the conclusions at the end of the discussion section as suggested, trying to be more consistent according to our findings. However, we can modify it again if it is considered necessary.
Round 2
Reviewer 2 Report
Thank you for improving your manuscript.